# Enhancing Human Activity Recognition with Siamese Networks: A Comparative Study of Contrastive and Triplet Learning Approaches

Byung-Rae Cha [1] and Binod Vaidya [2,*]

1 Information Technology Research Center, Chosun University, Gwangju 61452, Republic of Korea; brcha@smartx.kr
2 School of Electrical Engineering and Computer Science, University of Ottawa, Ottawa, ON K1N 6N5, Canada
* Correspondence: bnvaidya@mail.com; Tel.: +1-6139006635

**Abstract:** This paper delves into the realm of human activity recognition (HAR) by leveraging the capabilities of Siamese neural networks (SNNs), focusing on the comparative effectiveness of contrastive and triplet learning approaches. Against the backdrop of HAR's growing importance in healthcare, sports, and smart environments, the need for advanced models capable of accurately recognizing and classifying complex human activities has become paramount. Addressing this, we have introduced a Siamese network architecture integrated with convolutional neural networks (CNNs) for spatial feature extraction, bidirectional LSTM (Bi-LSTM) for temporal dependency capture, and attention mechanisms to prioritize salient features. Employing both contrastive and triplet loss functions, we meticulously analyze the impact of these learning approaches on the network's ability to generate discriminative embeddings for HAR tasks. Through extensive experimentation, the study reveals that Siamese networks, particularly those utilizing triplet loss functions, demonstrate superior performance in activity recognition accuracy and F1 scores compared with baseline deep learning models. The inclusion of a stacking meta-classifier further amplifies classification efficacy, showcasing the robustness and adaptability of our proposed model. Conclusively, our findings underscore the potential of Siamese networks with advanced learning paradigms in enhancing HAR systems, paving the way for future research in model optimization and application expansion.

**Keywords:** human activity recognition; Siamese neural networks; convolutional neural network; bidirectional LSTM; attention mechanism; contrastive loss function; triplet loss function



## 1. Introduction

Human activity recognition (HAR) is an interdisciplinary field of research that has garnered significant attention due to its wide range of applications, from healthcare monitoring and elderly care to sports analytics, interactive gaming, and smart environments. The core objective of HAR is to identify and classify different human activities based on data captured from various sensors, such as accelerometers, gyroscopes, or even video cameras. In healthcare, for example, HAR can facilitate continuous patient monitoring without the need for intrusive supervision, enabling early detection of deteriorating conditions or fall detection for the elderly. In smart homes, HAR can enhance energy efficiency and security by adapting the environment to the occupants' activities.

Traditionally, HAR has relied heavily on handcrafted feature extraction techniques, where domain expertise is used to select features from sensor data that are most indicative of different activities. These features are then fed into classical machine learning models like support vector machines (SVMs), decision trees, or K-nearest neighbors (KNNs) for classification. While effective in controlled settings, these traditional methods often fall short in more complex, real-world scenarios due to their inability to capture the high-dimensional and dynamic nature of human activities. Moreover, the manual feature

selection process is labor-intensive and may not generalize well across different tasks or datasets.

The advent of deep learning has provided new avenues for HAR, with models like convolutional neural networks (CNNs) and recurrent neural networks (RNNs), including their long short-term memory (LSTM) variants, demonstrating the ability to automatically learn representative features from raw data. These models have shown remarkable success in capturing the spatial and temporal complexities of human activities. However, deep learning approaches require substantial labeled data for training and can be computationally intensive, limiting their applicability in resource-constrained environments. Additionally, the black-box nature of these models often leads to challenges in interpretability, an important aspect in sensitive applications like healthcare.

Siamese neural networks represent a novel paradigm in the landscape of HAR, offering a way to mitigate some of the limitations associated with traditional and deep learning methods. By design, Siamese neural networks are adept at learning from relative comparisons rather than absolute label assignments, making them particularly suitable for tasks where the objective is to gauge similarity or dissimilarity between pairs or groups of data points, such as in HAR. The incorporation of CNN layers within the Siamese architecture allows for effective spatial feature extraction from sensor data, capturing intricate patterns and textures that characterize different activities. Following the CNN layers, Bi-LSTM networks are employed to model the temporal dependencies in the data, leveraging their ability to remember past information for extended periods and process sequences in both forward and backward directions. This bidirectional processing is crucial for understanding the sequential nature of human movements. To further enhance the model's focus on relevant features, attention mechanisms are integrated, enabling the network to dynamically weigh the importance of different inputs, thereby prioritizing features that are most salient for activity recognition.

Despite these advancements, the problem of robust feature representation and classification in HAR remains a significant challenge. The variability in human activities, coupled with differences in sensor placement, noise, and individual user characteristics, necessitates the development of more advanced techniques that can adapt to these complexities while maintaining high accuracy and efficiency.

This study delves into Siamese networks for HAR, with a focus on both contrastive and triplet learning methods. Our approach integrates Siamese neural networks (SNNs) with attention mechanisms tailored specifically for HAR—a combination not extensively explored in previous studies. We also critically analyze the performance of contrastive and triplet loss functions within this framework. Contrastive learning aims to refine embeddings by pulling similar instances together and pushing dissimilar ones apart, facilitated by contrastive loss functions. This enhances the network's ability to differentiate activities. Triplet learning extends this concept by using triplets (anchor, positive, and negative) to further optimize the embedding space, ensuring even more distinct separations between activities. The research also explores the use of stacking meta-classifiers, which combine predictions from various models to improve overall classification accuracy. The comparative analysis of these methodologies is intended to provide valuable insights into creating more efficient and interpretable HAR systems, demonstrating the versatility and potential of SNNs in advancing HAR applications.

## 2. Related Works

Human activity recognition (HAR) has seen significant advancements due to its applications in health, sports, and smart environments. Deep learning models have particularly enhanced HAR by effectively extracting spatial and temporal features from sensor data.

Traditional machine learning methods in HAR relied heavily on manual feature extraction, limiting generalization. Deep learning has revolutionized HAR by enabling automatic feature extraction, overcoming the limitations of traditional manual methods [1,2].

Wang et al.'s survey on deep learning for sensor-based activity recognition encapsulates this transition, highlighting deep learning's capability to enhance generalization performance and its adaptability to unsupervised and incremental learning tasks [1], while Tee et al. surveyed deep learning models, highlighting the success of hybrid systems that combine CNN and LSTM layers for activity recognition [2].

CNNs have gained prominence in HAR, due to their proficiency in pattern recognition and automatic feature extraction. Raj and Kos [3] delve into CNN's application in HAR, emphasizing its utility in interpreting temporal and spatial data from wearable sensors. Their study introduces a CNN-based model for classifying human activities, demonstrating a notable accuracy improvement on the WISDM dataset for HAR tasks.

The shift towards deep learning has become widely popular in recent years. Xia et al. [4] demonstrated leveraging LSTM and CNN architectures to automatically extract features from sensor data, significantly improving model performance and robustness across various datasets. Luwe et al. [5] presented a hybrid model combining 1D-CNN with Bi-LSTM, achieving remarkable accuracy on several datasets, and demonstrating the potential of deep learning in HAR, whereas Roobini and Naomi [6] explored the use of smartphones for HAR, employing ConvLSTM and RNNLSTM models to analyze sensor data, emphasizing the potential of deep learning in mobile-based HAR.

Furthermore, Zhao et al. [7] introduced a deep network using residual bidirectional LSTM, showcasing significant improvements in HAR accuracy, significantly on standard datasets, by leveraging both spatial and temporal dimensions. Li and Wang [8] developed a model combining residual networks and Bi-LSTM to capture complex activity patterns with high accuracy across several datasets.

Nafea et al. [9] introduced a novel methodology utilizing CNNs with varying kernel dimensions and Bi-LSTM for capturing features from sensor data, demonstrating improvements in HAR accuracy. Khan et al. [10] presented a hybrid CNN-LSTM model tailored for HAR, which excelled in extracting spatial and temporal features, marking a significant advancement in activity recognition research.

Siamese networks, known for their pairwise comparison approach, have been adapted for HAR to refine feature embeddings. Their ability to learn from relative comparisons makes them suitable for HAR tasks, as explored in various studies focused on enhancing model performance through advanced embedding techniques. For instance, Sheng and Huber [11] explored Siamese networks for weakly supervised HAR, utilizing similarity loss to train models that effectively segment and recognize continuous activity sequences without explicit labels. Contrastive and triplet learning have revolutionized HAR by refining embedding spaces for better activity differentiation. Li et al. [12] proposed similarity embedding networks trained with pairwise similarity loss, showcasing robustness against mislabeled samples and enhanced classification accuracy.

The literature underscores a transition from manual feature extraction to automatic feature learning through deep neural networks, significantly advancing HAR. The combination of spatial feature extraction via CNNs with temporal modeling using LSTM or Bi-LSTM layers has become a cornerstone in HAR models. Siamese networks, with their ability to learn from relative comparisons, present a promising avenue for HAR, especially when coupled with contrastive or triplet learning approaches. These methodologies not only improve the granularity of feature embeddings but also enhance a model's ability to generalize across diverse and noisy datasets.

## 3. Preliminaries

### 3.1. Basic Concepts of Convolutional Neural Networks (CNNs)

Convolutional neural networks (CNNs) stand at the forefront of feature extraction, particularly from visual data. CNNs employ layers of convolutions that apply various filters to the input data, effectively capturing spatial hierarchies of features. This process allows CNNs to learn complex patterns, from simple edges in the initial layers to intricate

details in deeper layers, making them ideal for tasks requiring detailed feature analysis, such as image and signal processing. Figure 1 shows an overview of the CNN.

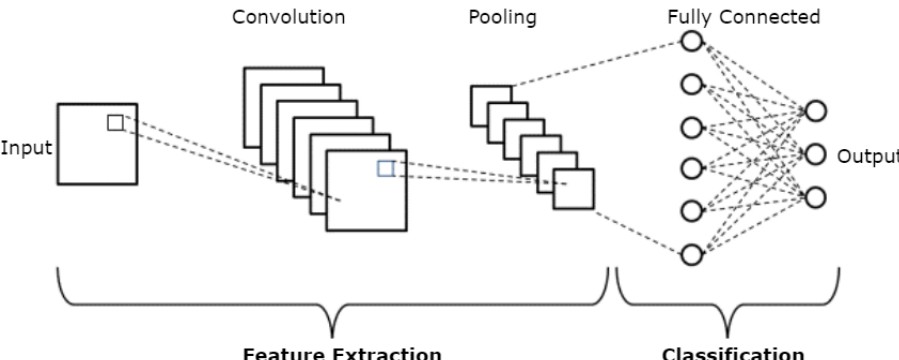

**Figure 1.** Overview of CNN.

### 3.2. Fundamentals of Bidirectional Long Short-Term Memory (Bi-LSTM) Networks

Bidirectional long short-term memory (Bi-LSTM) networks extend the capabilities of traditional recurrent neural networks (RNNs) by processing data points in both forward and backward directions. This bidirectionality enables the network to preserve information from both past and future states, offering a more comprehensive understanding of temporal dynamics. Bi-LSTMs are particularly effective in time-series data applications, such as speech and language processing, where the context from both directions is crucial for accurate interpretation. Figure 2 shows an overview of the Bi-LSTM network.

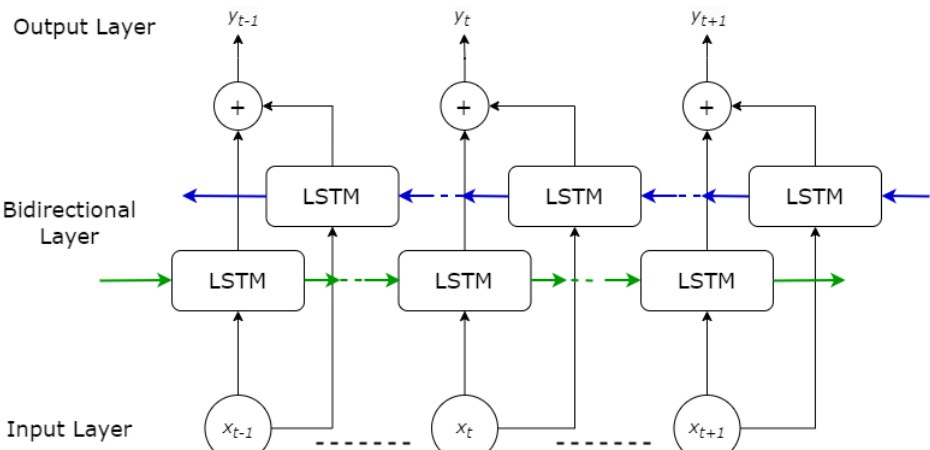

**Figure 2.** Overview of Bi-LSTM.

### 3.3. Overview of Attention Mechanisms

Attention mechanisms have revolutionized the way neural networks process sequences, allowing models to focus on specific parts of the input data that are more relevant to the task at hand. This focus is achieved by assigning different weights to various parts of the data, enabling models to prioritize and aggregate information from the most pertinent areas. Attention mechanisms enhance model performance in tasks requiring nuanced understanding and contextual awareness, such as machine translation and document summarization. Figure 3 shows an overview of the attention mechanism.

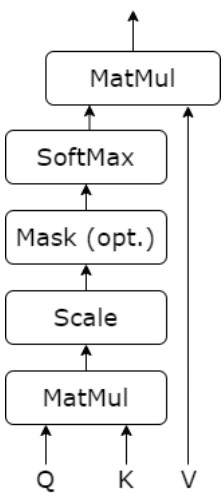

**Figure 3.** Overview of attention mechanisms.

The attention mechanism utilized in our SNNs for HAR is a critical component that helps the model prioritize and consolidate information from input sequences, thereby enhancing its ability to discern subtle activity patterns.

In the model, the attention mechanism is designed to allow the network to focus on the key input elements, query (Q), key (K), and value (V). The query (Q) component represents the current state or element at a specific timestep in the sequence, seeking relevant information from the entire input; keys (K) unlock information in value elements by being compared against queries to determine their relevance or importance; and values (V) contain the actual information from the input data, providing content for the model to focus on once their corresponding keys are deemed relevant through comparison with a query.

The operations involving Q, K, and V are depicted as follows.

1. Dimension matching: initially, queries and keys are transformed through dense layers to ensure they match in dimensionality. This transformation facilitates the effective computation of attention scores.

2. Score computation: attention scores are computed by taking the dot product of the query matrix, $Q$, with the transpose of the key matrix, $K$, mathematically represented as scores $= QK^T$, indicating the relevance of each input element.

3. Application of softmax: the softmax function normalizes the scores into a probability distribution, represented as attention weights $=$ softmax (scores), focusing the model's attention by amplifying the highest scores and diminishing the lower ones, ensuring the weights sum to 1 for interpretability as probabilities.

4. Weighted sum of values: the final step in the attention mechanism involves computing a weighted sum of values using the previously calculated attention probabilities to form a context vector, context vector $= \sum(\text{attention weights} \times V)$, dynamically aggregating the most relevant information for the model to focus on.

### 3.4. Overview of Siamese Networks, Contrastive Learning, and Triplet Learning

Siamese networks, renowned for their dual architecture, are adept at comparing and contrasting pairs of input data. By sharing weights between identical subnetworks, they excel in tasks that require assessing the similarity or relationship between two data points, such as in verification and matching tasks. In the Siamese network architecture, the principle of weight sharing is crucial and particularly instrumental when it comes to tasks that involve learning fine distinctions between similar and dissimilar instances.

In a Siamese network, each identical subnetwork processes different elements of input pairs or triplets with shared weights, ensuring consistent feature extraction across all inputs. Uniform feature extraction across subnetworks in Siamese networks ensures fair

and consistent comparisons between different inputs, crucial for their primary function of effectively comparing and contrasting inputs.

Figure 4 depicts a generic Siamese network.

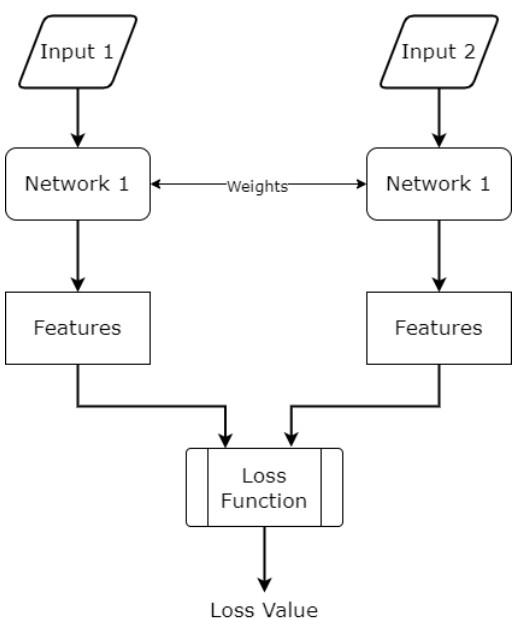

**Figure 4.** Generic Siamese network.

Contrastive learning, commonly integrated within Siamese frameworks, enhances this capability by encouraging the model to differentiate between similar (positive) and dissimilar (negative) pairs, thus cultivating robust and discriminative feature representations. In the case of contrastive learning, weight sharing ensures that the feature extraction process for both inputs in a pair is symmetrical, allowing the network to accurately measure similarities or differences based on the same feature representation.

Expanding on this foundation, triplet learning introduces an advanced perspective by utilizing triplets of data points, consisting of an anchor, a positive example (similar to the anchor), and a negative example (dissimilar to the anchor). The triplet loss function, a key component of triplet learning, aims to ensure that the anchor is closer to the positive example than to the negative example by a defined margin. This approach not only distinguishes between similar and dissimilar pairs but also fine-tunes the relative distances within the embedding space, fostering even more nuanced feature representations. In triplet learning, weight sharing across the branches processing the anchor, positive, and negative inputs guarantees that the distance metrics are computed consistently.

Sharing weights in a network streamlines learning by reducing the number of parameters, speeding up training, and lessening computational demands. Using the same weights for different inputs helps the network generalize better, a key advantage in applications like HAR where consistent performance across various conditions and subjects is crucial. Consistent feature extraction ensures that embeddings are comparable, enhancing the effectiveness of contrastive and triplet loss functions that depend on reliable distances between embeddings.

## 4. Proposed System Architecture

The proposed system architecture for human activity recognition (HAR) using a Siamese neural network incorporates a synergistic combination of convolutional neural networks (CNNs), bidirectional long short-term memory (Bi-LSTM) units, and attention mechanisms. The system is further enhanced with supervised contrastive learning, employing contrastive and triplet loss functions, and is finalized with a stacking meta-classifier for robust activity classification. This architecture is designed to leverage the strengths of each

component, resulting in a robust and efficient system capable of effectively tackling the challenges of spatial and temporal variability in human activities from smartphone sensor data, ensuring high accuracy and reliability in activity recognition.

### 4.1. Siamese Network Architecture

The proposed Siamese network for human activity recognition (HAR) is designed to harness the strengths of convolutional neural networks (CNNs), bidirectional long short-term memory (Bi-LSTM) networks, and attention mechanisms. This integration facilitates a robust extraction and representation of features from sensor data, capturing both spatial patterns and temporal dynamics inherent in human activities. Figure 5 illustrates the proposed Siamese neural network architecture with a contrastive loss function and a stacking meta-classifier.

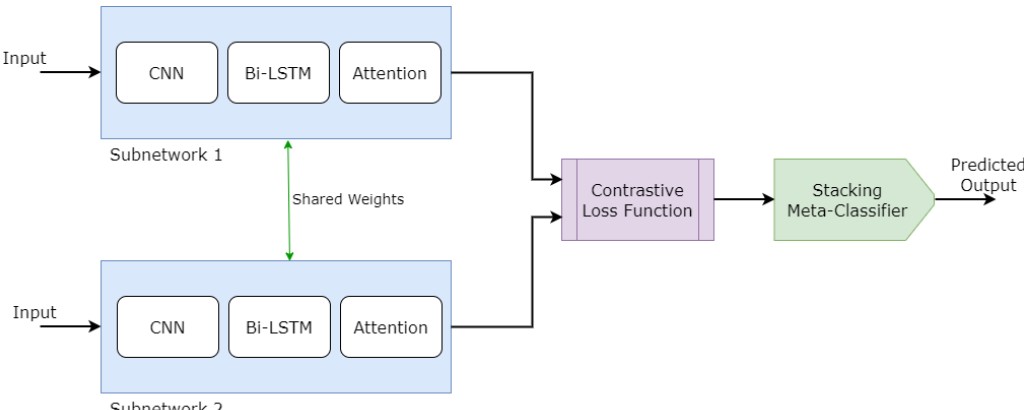

**Figure 5.** Proposed Siamese neural network architecture with contrastive loss function and stacking meta-classifier.

#### 4.1.1. Feature Extraction and Representation

The process begins with the Siamese network components performing their respective roles in feature extraction and representation.

#### A. CNN:

The CNN component of the Siamese network is pivotal in extracting spatial features from the input sensor data. The architecture typically consists of multiple convolutional layers followed by pooling layers. Each convolutional layer, $C_i$, applies a set of learnable filters, $k$, followed by a nonlinear activation function, $\sigma$, typically ReLU($\sigma(x) = max(0, x)$), to introduce nonlinearity. The operation can be represented as:

$$C_i(x) = \sigma(W_i * x + b_i) \tag{1}$$

where $W_i$ and $b_i$ denote the weights and bias for the $i$th layer, and $*$ represents the convolution operation.

Pooling layers (usually max pooling) follow convolutional layers to reduce the spatial dimensions of the feature maps, thereby decreasing the computational complexity and enhancing the network's ability to generalize by focusing on dominant features.

The initial layers capture basic features, which are combined into more complex representations in deeper layers. This hierarchical feature extraction process is crucial for identifying the nuanced spatial characteristics of different activities, making CNNs an integral part of the Siamese network's architecture for HAR.

#### B. Bi-LSTM:

Following the CNN layers, Bi-LSTM networks are employed to capture the temporal dependencies in the activity data. Unlike traditional unidirectional LSTMs that process

data in a single direction, Bi-LSTMs analyze the sequence in both forward and backward directions, providing a comprehensive context of the sequence at each point. This bidirectional analysis is particularly beneficial for HAR, where the context before and after a particular moment can significantly influence the interpretation of an activity.

Bi-LSTMs consist of two LSTM layers that process the data in opposite directions, and their outputs are combined at each time step. For a given time step, *t*, the forward LSTM ($\overrightarrow{LSTM}$) and backward LSTM ($\overleftarrow{LSTM}$) operations are defined as:

$$\overrightarrow{h} = \overrightarrow{LSTM}\left(x_t, \overrightarrow{h_{t-1}}\right)$$
$$\overleftarrow{h} = \overleftarrow{LSTM}\left(x_t, \overleftarrow{h_{t-1}}\right) \tag{2}$$

The final output at each time step $h_t$ is the concatenation of forward, $\overrightarrow{h_t}$, and backward, $\overleftarrow{h_t}$, hidden states, capturing information from both past and future contexts.

This setup allows the network to retain information from both past and future states, capturing the dynamic temporal patterns and transitions between different states of activity, which are critical for accurate recognition.

C. Attention Mechanism:

An attention mechanism is applied to the outputs of the Bi-LSTM layers, enabling the network to focus on the most relevant temporal features for activity recognition.

The attention mechanism assigns weights, $\alpha_t$, to the LSTM outputs, $h_t$, focusing the network on the most relevant temporal features. The context vector, *c*, is computed as follows:

$$e_t = a(h_t)$$
$$\alpha_t = \frac{exp(e_t)}{\sum_{t=1}^{T} exp(e_t)} \tag{3}$$
$$c = \sum_{t=1}^{T} \alpha_t h_t$$

where $a(\cdot)$ is a learnable function that computes the relevance score, $e_t$, of each time step, and *T* is the total number of time steps.

It can be observed that the relevance score, $e_t$, is computed. Here, the function, *a*, transforms the hidden state, $h_t$, at each time step, *t*, to compute a relevance score, $e_t$, determining the contribution of each time step's data to the final output based on its assessed relevance. The softmax function applied to relevance scores, $e_t$, uses the exponential function to amplify score differences and normalizes these scores into attention weights, $\alpha_t$, representing probabilities that reflect the importance of each time step's data. Finally, the context vector, *c*, is calculated as the weighted sum of hidden states, $h_t$, using attention weights, $\alpha_t$, synthesizing input data by emphasizing features from the most relevant time steps.

This mechanism dynamically assigns higher weights to more informative features, enhancing the network's ability to discern subtle differences between similar activities and improving the overall accuracy of the HAR system. Additionally, by learning the function, *a*, and relevance scores, $e_t$, the model adapts to varying data and contexts without manual intervention, while the attention weights, $\alpha_t$, improve interpretability by revealing which data parts are deemed important.

4.1.2. Embedding Generation in Siamese Network

Within the Siamese architecture, the processed features from the CNN, Bi-LSTM, and attention layers are combined to generate a comprehensive embedding for each input sample. This embedding serves as a compact and informative representation of the sensor data, suitable for distinguishing between various activities based on their inherent spatial

and temporal characteristics. The Siamese network leverages these embeddings to compare input samples, learning a metric space where similar activities are closely positioned, and dissimilar ones are distant. This embedding generation process is central to the network's ability to accurately recognize and classify human activities.

4.1.3. Siamese Network with Contrastive and Triplet Learning Approaches

A Siamese network architecture is characterized by twin branches that process input data in parallel, with each branch handling one element of an input pair or triplet. These branches are identical, sharing the same weights and architecture, which ensures uniform feature extraction across the inputs. This shared-weight design is fundamental for the network's capability to assess and differentiate the extracted features from each input, making it particularly effective for tasks that involve comparing similarities or discrepancies between data samples.

In the context of HAR, the Siamese network leverages contrastive and triplet learning approaches to refine its ability to distinguish between similar and dissimilar activity instances. These learning paradigms are underpinned by specially designed loss functions that guide the network towards generating embeddings where similar instances are closer together, and dissimilar instances are further apart in the embedding space.

A. Contrastive Loss:

The contrastive loss function is pivotal in the contrastive learning approach, primarily focusing on pairs of input samples, and is used to differentiate between positive (similar) and negative (dissimilar) pairs.

For a pair of embeddings, $x_i$ and $x_j$, the contrastive loss, $L_{contr}$, can be mathematically represented as:

$$L_{contr}(x_i, x_j, Y) = \frac{1}{2}(1 - Y)(D(x_i, x_j))^2 + \frac{1}{2}(Y)(max(0, m - D(x_i, x_j)))^2 \qquad (4)$$

where:

$Y$ is a binary label indicating whether $x_i$ and $x_j$ are similar ($Y = 0$) or dissimilar ($Y = 1$).

$D(x_i, x_j)$ is the distance between the embeddings of the $i$th and $j$th samples.

$m$ represents the margin, a hyperparameter that defines how far apart the dissimilar pairs should be.

This function penalizes the network for close embeddings of dissimilar pairs and distant embeddings of similar pairs, thereby encouraging similar activities to have closer embeddings and vice versa.

B. Triplet Loss:

The triplet loss function extends the contrastive learning approach by considering triplets of samples, consisting of an anchor, $a$, a positive example, $p$ (similar to the anchor), and a negative example, $n$ (dissimilar to the anchor). The triplet loss, $L_{trip}$, is defined as:

$$L_{trip}(a, p, n) = max(0, D(a, p) - D(a, n) + m) \qquad (5)$$

where:

$D(a, p)$ and $D(a, n)$ are the distances between the anchor and the positive example, and the anchor and the negative example, respectively.

$m$ is the margin, enforcing a minimum separation between the positive and negative pairs relative to the anchor.

The triplet loss aims to ensure that the anchor is closer to the positive example than to the negative example by at least margin $m$, thereby structuring the embedding space in a way that reflects the relationships within the input data.

Through these loss functions, the Siamese network, whether in its twin or triplet configuration, optimizes its embeddings to effectively capture the nuances of human activities, facilitating accurate recognition and classification.

### 4.2. Stacking Meta-Classifier as a Decision Maker

The final component of the proposed system is the stacking meta-classifier, which has decision-making capabilities and integrates the outputs from the Siamese network to perform the final classification of activities. This meta-classifier combines predictions from multiple base classifiers, each trained on the embeddings generated by the Siamese network, to produce a final prediction. The stacking approach leverages the diversity of the base classifiers, allowing for a more nuanced and accurate classification decision. Figure 6 depicts a block diagram representation of a stacking meta-classifier.

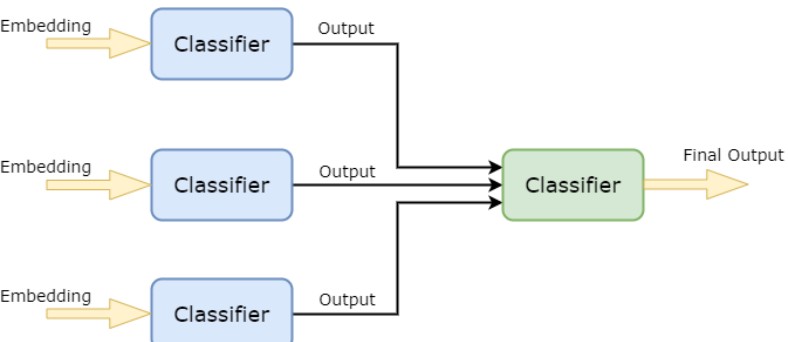

**Figure 6.** Block diagram representation of stacking meta-classifier.

The incorporation of the stacking meta-classifier provides several benefits, including:

- Robustness against overfitting by combining multiple models.
- Reducing the likelihood of the final classifier being overly reliant on the idiosyncrasies of the training data.
- Improved predictive performance, as it can capture complex patterns that may be missed by individual classifiers.

## 5. Implementation Aspects

This section discusses the dataset and preprocessing.

### 5.1. Dataset

For the implementation of the proposed Siamese network with contrastive and triplet learning approaches for human activity recognition (HAR), the widely recognized UCI HAR dataset is utilized. This dataset is a benchmark in the field of HAR and provides a comprehensive foundation for developing and evaluating machine learning models.

#### 5.1.1. UCI HAR Dataset Overview

The UCI HAR dataset was collected from a group of 30 volunteers within an age bracket of 19–48 years. Each participant performed six activities: walking, walking upstairs, walking downstairs, sitting, standing, and laying. The activities were recorded using a smartphone worn on the waist, equipped with embedded inertial sensors.

Sensor Types and Data Characteristics

- Accelerometer and gyroscope: the dataset incorporates readings from the smartphone's accelerometer and gyroscope. The accelerometer captures linear acceleration, while the gyroscope measures angular velocity. Both sensors provide three-dimensional data, representing the x-, y-, and z-axes.

- Sampling rate: the signals were sampled in fixed-width sliding windows of 2.56 s, with a 50% overlap between consecutive windows, resulting in a sampling rate of 50 Hz.
- Signal processing: the signals from both sensors were preprocessed by applying noise filters and then sampled in fixed-width sliding windows. The sensor acceleration signal, which has gravitational and body motion components, was separated using a Butterworth low-pass filter into body acceleration and gravity.
- Feature extraction: from each window, a vector of features was obtained by calculating variables from the time and frequency domain. This includes measures such as mean, standard deviation, median, max, min, skewness, kurtosis, and various others, leading to 561 feature vectors per sample.

The dataset was divided into two sets: 70% of the volunteers were selected to generate the training data and 30% the test data. Figure 7 visually represents the distribution of observations across different activities in a UCI HAR. Such a distribution for total data is shown in Figure 7a; similarly, training data and test data are shown in Figure 7b and Figure 7c, respectively.

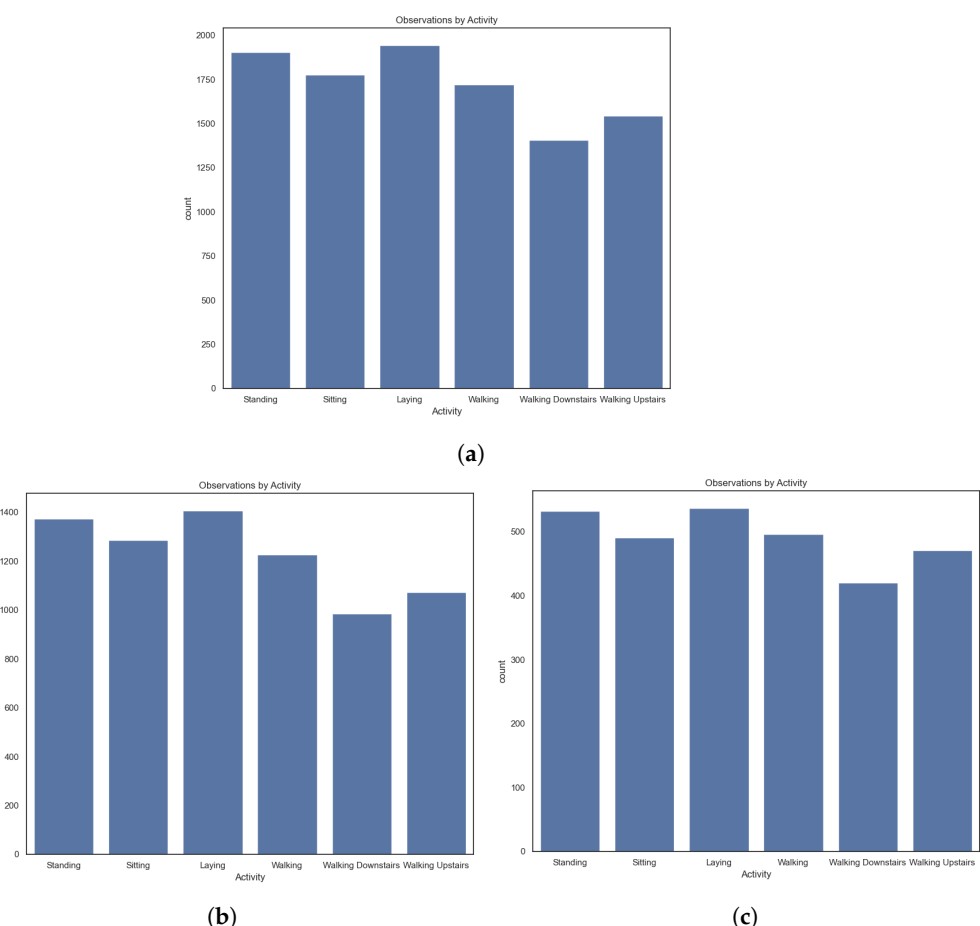

**Figure 7.** Distribution of observations by activity (**a**) Total data. (**b**) Training data. (**c**) Test data.

This richly annotated dataset not only provides the raw time-series data but also the preprocessed features, making it versatile for different kinds of analysis and model training approaches.

For the proposed Siamese network, both raw sensor data for deep feature extraction and preprocessed features for baseline comparisons can be employed. The diversity of activities, coupled with the multi-dimensional sensor data, presents an excellent opportunity to explore and validate the efficacy of the Siamese network architecture, contrastive learning, and triplet loss functions in distinguishing complex human activities.

5.1.2. Data Preprocessing

The data preprocessing stage is crucial for preparing the UCI HAR dataset for effective learning by the Siamese network. The process involves several steps:

- Normalization: each feature in the dataset is normalized to have a zero mean and unit variance, ensuring consistency across different scales of sensor readings.
- Windowing: the continuous sensor data stream is segmented into fixed-sized windows with a 50% overlap, as per the original dataset's configuration. Each window is treated as an independent sample.

*5.2. Experimental Setup*

5.2.1. Configuration of Neural Network Models

The configuration of the Siamese network's subnetwork involves a meticulously designed architecture that leverages a combination of convolutional neural networks (CNNs), bidirectional long short-term memory (Bi-LSTM) units, and a custom attention mechanism to process and analyze sensor data for human activity recognition (HAR).

A. Initial Feature Extraction with CNN Layers:

The model begins with an input layer that accommodates the preprocessed sensor data, followed by a Conv1D layer with a defined number of filters and kernel size, using 'same' padding to maintain the input's dimensionality while extracting spatial features. Batch normalization is applied to normalize the activations, followed by a ReLU activation function for nonlinearity. A MaxPooling1D layer with a pool size of 2 reduces the dimensionality of the features, which is essential for the network's efficiency and to capture the most relevant spatial features.

This setup is duplicated with another Conv1D layer, where the number of filters is adjusted to refine the feature maps. The sequence of batch normalization, ReLU activation, and max pooling is repeated to further process the spatial features for temporal analysis.

B. Temporal Dependency Modeling with Bi-LSTM Layers:

The spatial features are then passed through Bi-LSTM layers to model the temporal dependencies within the data. The first Bi-LSTM layer, with a set number of units, incorporates dropout to mitigate overfitting and processes the data bidirectionally, capturing temporal dependencies from past and future contexts. A ReLU activation function is applied to ensure nonlinearity. A second Bi-LSTM layer with a similar setup refines the temporal feature representation.

C. Feature Prioritization with Attention Mechanism:

A custom attention mechanism is employed on the outputs of the Bi-LSTM layers to prioritize and focus on the most relevant temporal features for activity recognition. This mechanism calculates attention weights for the Bi-LSTM outputs, emphasizing significant features. A Lambda layer is used to extract key elements from the attention-weighted sequence, focusing on the most informative temporal features for the final activity classification.

5.2.2. Stacking Meta-Classifier for Final Classification

A stacking meta-classifier is employed for the final classification, integrating outputs from various base classifiers trained on the optimized embeddings. The configuration of each base classifier, such as decision trees, extra trees, random forest, and LightGBM, is specified with parameters tailored to the HAR task. The final classification is performed by an SVC classifier, which integrates the base classifiers' predictions to produce the final activity labels.

### 5.2.3. Training Procedure

The training process for the Siamese neural network in human activity recognition (HAR) is meticulously designed to ensure effective learning and generalization. The procedure encompasses optimization strategies, hyperparameter tuning, and validation approaches, tailored to the specific requirements of the network architecture and the HAR task.

Case 1: Training with Contrastive Loss Function:

For the contrastive learning approach, the network is trained on pairs of input samples, using the contrastive loss function to minimize the distance between embeddings of similar activities and maximize the distance for dissimilar activities. This training regime encourages the model to learn embeddings that cluster similar activities closely in the embedding space, enhancing the discriminative capability of the network.

Case 2: Training with Triplet Loss Function:

In the triplet learning scenario, the network is trained on triplets of samples (anchor, positive, and negative), utilizing the triplet loss function. This approach aims to ensure that the anchor is closer to the positive sample than to the negative sample by a defined margin, further structuring the embedding space for improved activity recognition.

### 5.2.4. Optimization and Hyperparameters

The training utilizes both Adam and stochastic gradient descent (SGD) with momentum optimizers, leveraging their unique advantages:

- Adam optimizer: known for its adaptive learning rate adjustment, the Adam optimizer is initialized with a learning rate of 0.0001, beta1 set to 0.96, beta2 to 0.99985, and epsilon to $1 \times 10^{-8}$. This configuration helps in accommodating the variances in gradients, making Adam particularly effective for datasets with noisy or sparse gradients.
- SGD with momentum: SGD, augmented with a momentum term of 0.6, is employed to expedite the training by smoothing the gradient descent process. The momentum assists in overcoming potential oscillations and stabilizes the updates, facilitating a more direct path toward the optimization minima.

Key hyperparameters are carefully selected to optimize the network's performance:

- Regularization: L2 regularization, with a weight decay of $1 \times 10^{-3}$ is applied to penalize large weights, reducing the risk of overfitting.
- Dropout: a dropout rate of 0.5 is implemented in the LSTM layers to randomly exclude a subset of features during training, further preventing the model from becoming overly dependent on specific neurons.

Hyperparameter tuning was conducted iteratively, with the validation set performance guiding the adjustments. This involves experimenting with various hyperparameter combinations to find the optimal setup that maximizes validation accuracy while maintaining a balance between learning efficiency, model complexity, and generalization.

The final tuning of hyperparameters in our model encompasses:

— Batch size: we experimented with different batch sizes to evaluate their effects on model generalization and computational efficiency during training. After testing batch sizes of 32 and 64, it became evident that a batch size of 64 strikes an optimal balance, enhancing both the training speed and overall model performance.
— Number of epochs: the total number of epochs was determined by observing the behavior of the loss function during training. We tested the model across 50, 100, and 200 epochs, with an early stopping mechanism triggered if the validation loss did not improve for 10 consecutive epochs. Our findings suggest that extending training to 200 epochs generally allows the model sufficient time to converge to a robust solution.

To visually monitor the training progress and facilitate timely adjustments, we plotted loss function graphs for both the training and validation phases across different epochs. These graphs are crucial for understanding how the loss evolves, identifying when the model begins to plateau, and determining the optimal moment for early stopping based on stabilization or deterioration in loss reduction. Figure 8 presents these loss function graphs: Figure 8a illustrates the training and validation loss over 100 epochs, while Figure 8b extends this analysis to 200 epochs, providing insights into the longer-term training dynamics.

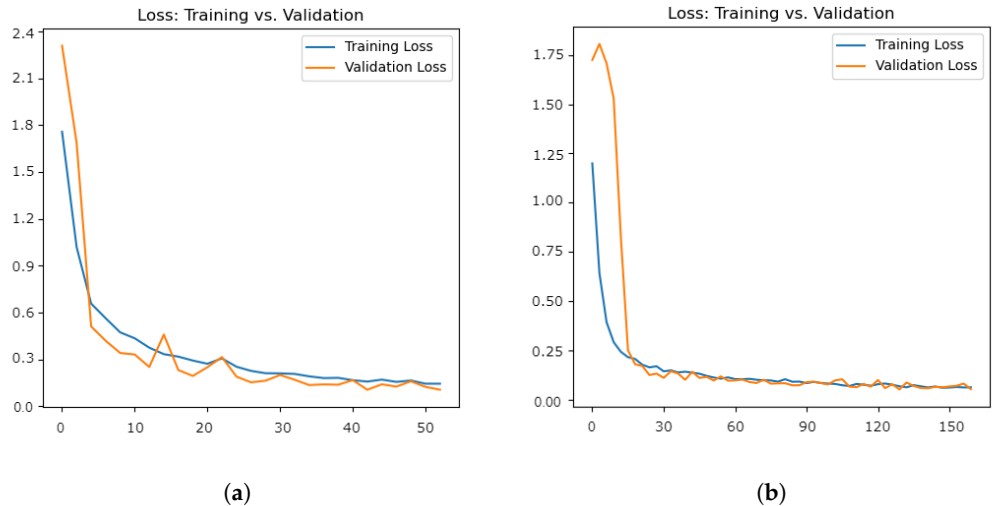

(**a**)                                                 (**b**)

**Figure 8.** Loss function graphs. (**a**) Training and validation loss over 100 epochs. (**b**) Training and validation loss over 200 epochs.

### 5.3. Results and Analysis

### 5.3.1. Overview of Performance Metrics

To comprehensively evaluate the performance of the proposed Siamese network model for human activity recognition (HAR), we employ several standard evaluation metrics, namely accuracy, precision, recall, and F1 score. These metrics provide insights into various aspects of model performance, from general accuracy to the balance between precision and recall.

A. Accuracy:

This metric represents the overall correctness of the model, calculated as the ratio of correctly predicted observations to the total observations. It gives a quick overview of the model's effectiveness but may not always provide a complete picture, especially in imbalanced datasets.

B. Precision:

Precision indicates the ratio of correctly predicted positive observations to the total predicted positives. High precision relates to a low false positive rate, crucial in scenarios where the cost of a false positive is high.

C. Recall (Sensitivity):

Recall measures the ratio of correctly predicted positive observations to all observations in the actual class. It is particularly important in cases where missing a positive (e.g., failing to recognize an activity) is critical.

D. F1 score:

The F1 score is the weighted average of precision and recall, taking both false positives and false negatives into account. It is a better measure than accuracy for imbalanced datasets. Initially, we considered both the mean and the weighted F1 scores. While calculat-

ing the mean F1 score, which averages the F1 scores for each class, the model's effectiveness tends to distort—overestimating performance on prevalent activities while undervaluing it on rarer ones. In contrast, the weighted F1 score compensates for this imbalance by weighting the score of each class according to its frequency, thus providing a more accurate reflection of the model's performance across diverse activities. Ultimately, we decided to use the weighted F1 score only for this study. This metric is particularly apt for HAR tasks where some activities might be overrepresented and others underrepresented in the dataset. Using the weighted F1 score ensures a fairer assessment of the model's effectiveness across all activity types, aligning with our aim to develop a robust HAR system.

E. Confusion Matrix:

The confusion matrix provides a detailed breakdown of the model's performance across different classes, showing the true positives, false positives, true negatives, and false negatives for each activity class. This can help identify specific classes where the model may be underperforming.

By leveraging these metrics, we can gain a comprehensive understanding of the proposed Siamese network model's strengths and limitations, guiding further improvements and refinements for enhanced HAR performance.

5.3.2. Results

The experimental results derived from training the Siamese network with both contrastive and triplet loss functions, using two different optimizers (SGD and Adam), provide insightful revelations about the model's performance and the impact of loss functions and optimization strategies on human activity recognition (HAR).

A. Contrastive Loss Function Results:

Upon applying the SGD optimizer, the model achieved an accuracy of 93.98%, with both precision and recall at 93.92%, leading to an F1 score of 93.9%. The alignment between precision and recall indicates a balanced performance in identifying true positives and minimizing false negatives and positives.

Using the Adam optimizer, the Siamese network exhibited marginally reduced performance metrics, achieving an accuracy of 92.48%, with precision and recall hovering around 92.45% and 92.45%, and a corresponding F1 score of 92.39%. This slight decrement in performance relative to the SGD optimizer could be ascribed to the adaptive learning rate mechanism of Adam. Although this feature typically aids in achieving faster convergence, it may have resulted in less than optimal progression through the specific loss landscape of this HAR task, slightly impacting the overall model efficacy.

The results from the contrastive loss function experiments suggest that, while both optimizers perform commendably, SGD offers a slight edge in this context, potentially due to its simpler and more consistent update mechanism, which might be better suited to the contrastive loss landscape in this specific HAR task.

B. Triplet Loss Function Results:

Applying the SGD optimizer with the triplet loss function, the model trained with the SGD optimizer showed a notable improvement, achieving an accuracy of 95.21%, precision at 95.19%, recall at 95.18%, and an F1 score of 95.18%. This enhancement in performance metrics, compared with contrastive loss, highlights the efficacy of the triplet loss function in structuring the embedding space in a way that more effectively discriminates between different activities.

In the case of the Adam optimizer, using the triplet loss function, the performance with Adam, while still robust, was slightly lower than with SGD, with accuracy at 93.2%, precision at 93.21%, recall at 93.16%, and an F1 score of 93.15%. This indicates that, while Adam's adaptive learning rate mechanism is effective, it may not fully capitalize on the structural advantages offered by the triplet loss function, as SGD does.

The superior results achieved with the triplet loss function, particularly with the SGD optimizer, underscore the benefit of utilizing a triplet framework for HAR tasks. The structured approach of comparing an anchor with both a positive and a negative sample likely provides a more nuanced and effective way to differentiate between activities, leading to improved model performance. Table 1 shows various evaluation metrics for SSN-CL and SSN-TL with Adam and SGD optimizers.

**Table 1.** Evaluation metrics for SSN-CL and SSN-TL with Adam and SGD optimizers.

| Loss Function | Contrastive | | Triplet | |
|---|---|---|---|---|
| Metrics/Optimizer | SGD | Adam | SGD | Adam |
| Accuracy | 93.98% | 92.48% | 95.21% | 93.2% |
| Precision | 93.92% | 92.45% | 95.19% | 93.21% |
| Recall | 93.92% | 92.43% | 95.18% | 93.16% |
| F1 score | 93.9% | 92.39% | 95.18% | 93.15% |

Figure 9 illustrates the overall accuracy for contrastive vs. triplet loss functions.

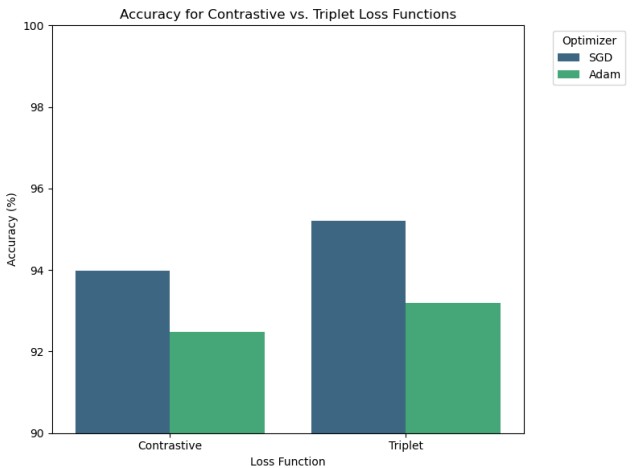

**Figure 9.** Accuracy for contrastive vs. triplet loss functions.

Figure 10 depicts the confusion matrix for the Siamese network with the contrastive loss function. Such confusion matrices for the Adam optimizer and SGD optimizer are depicted in Figure 10a and Figure 10b, respectively.

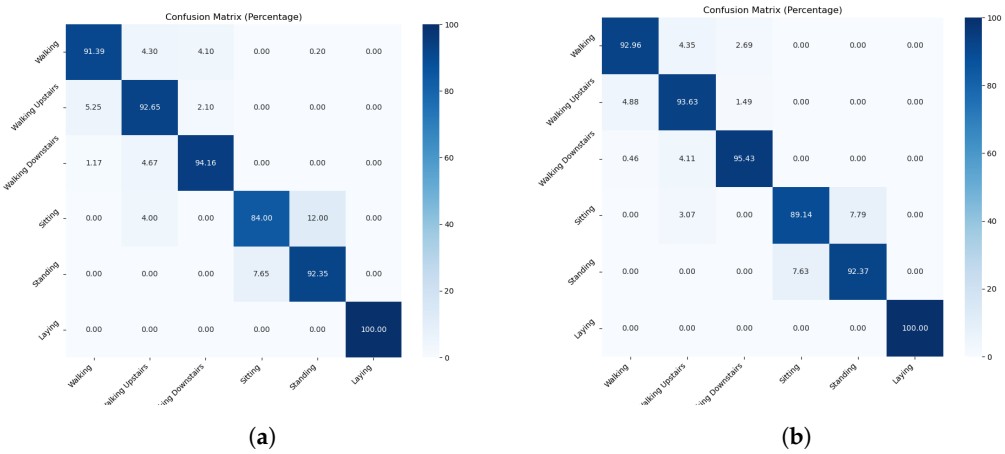

(**a**)          (**b**)

**Figure 10.** Siamese network with contrastive loss function. (**a**) Adam optimizer. (**b**) SGD optimizer.

Figure 11 depicts the confusion matrix for the Siamese network with the triplet loss function. Such confusion matrices for the Adam optimizer and SGD optimizer are depicted in Figure 11a and Figure 11b, respectively.

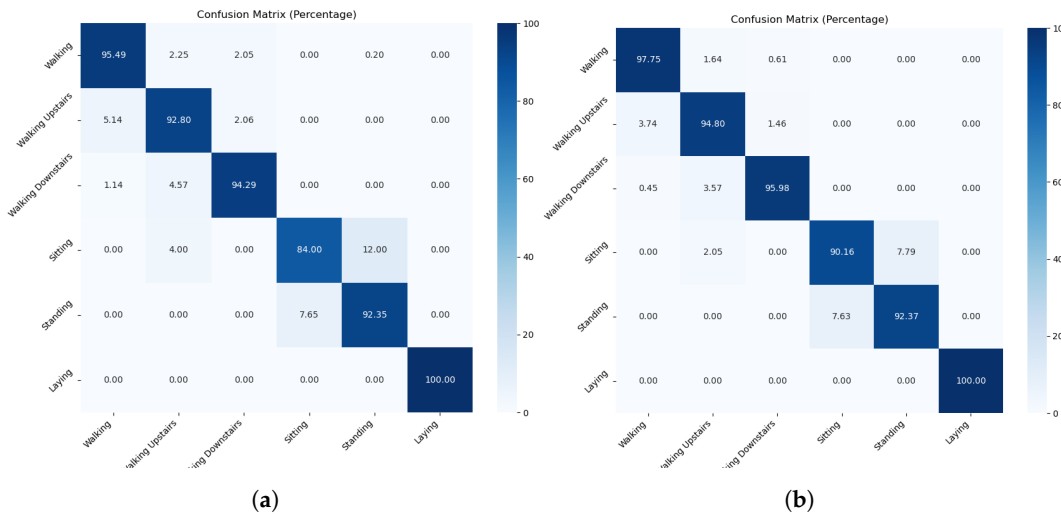

(**a**)                                                                         (**b**)

**Figure 11.** Siamese network with triplet loss function. (**a**) Adam optimizer. (**b**) SGD optimizer.

The discrepancy in the performance of Adam and SGD optimizers, particularly in classifying the "sitting" activity, arises from their inherent optimization behaviors. Adam uses adaptive learning rates tailored to each parameter based on gradient moments, which can sometimes lead to issues like overshooting in scenarios with subtle feature differences, as seen in the frequent misclassification of "sitting" as "standing". In contrast, SGD updates all parameters uniformly with a single rate, leading to more stable convergence and possibly better generalization in tasks with closely similar activities. Adam's approach might introduce biases due to its variable learning rates, affecting its ability to distinguish between similar physical states like "sitting" and "standing".

### 5.3.3. Benchmark Model Evaluation

To thoroughly assess the efficacy of our proposed Siamese neural network (SNN) approach, we constructed a benchmark model equipped with a CNN layer, a Bi-LSTM layer, and an attention mechanism. This configuration serves as a baseline to rigorously test the performance of our SNN under different training dynamics. We implemented training sessions using contrastive loss, focusing on evaluating the improvements this configuration brings to the model's discriminative power for human activity recognition (HAR) tasks. Such systematic training and evaluation highlight the SNN's capacity to significantly enhance feature distinction and recognition accuracy in complex pattern analysis scenarios, achieving an impressive accuracy of 91.5%. This analysis not only validates the robustness of our methodology but also demonstrates its potential to set new benchmarks in the field.

### 5.3.4. Results Comparison with Well-Known Baseline Models

The performance of the proposed Siamese neural network model, trained with contrastive (SSN-CL) and triplet loss (SSN-TL) functions, has been evaluated against several well-established baseline models in the domain of human activity recognition (HAR). The comparison is presented in terms of accuracy and F1 score, two key metrics that provide insights into the model's overall effectiveness and balance between precision and recall, respectively.

The table delineates the comparative analysis of various algorithms.

A. LSTM:

A standard LSTM model serves as a baseline to assess the importance of temporal feature modeling in HAR. Such an LSTM model provides an accuracy and F1 score of 90.8%. While effective for capturing temporal dependencies, its relatively simpler architecture might limit its capability to handle the complex spatial–temporal dynamics in HAR data.

B. Bi-LSTM:

The Bi-LSTM model, with its bidirectional processing of temporal data, can evaluate the benefits of capturing temporal dependencies from both past and future contexts. Bi-LSTM shows a slight improvement over the standard LSTM, achieving an accuracy and F1 score of 91.1%. This improvement underscores the value of capturing both past and future context in activity recognition.

C. Res-LSTM:

Incorporating residual connections, the residual LSTM (Res-LSTM) model further enhances the network's ability to learn from the data, reaching an accuracy of 91.6% and an F1 score of 91.5%. The residual connections likely aid in mitigating the vanishing gradient problem, allowing for deeper and more effective learning.

D. ConvLSTM:

The ConvLSTM model, which combines convolutional layers with LSTM units, achieves an accuracy of 92.24%. The absence of an F1 score makes it difficult to evaluate its precision–recall balance, but the accuracy suggests a significant advantage in capturing spatial features alongside temporal dependencies.

E. SSN-CL (Siamese Network with Contrastive Loss):

The Siamese network trained with a contrastive loss function demonstrates an accuracy of 93.98% and an F1 score of 93.9%, indicating a robust performance that benefits from the comparative learning framework and effective embedding of activity features.

F. SSN-TL (Siamese Network with Triplet Loss):

The most notable performance is observed with the Siamese network employing triplet loss, yielding the highest accuracy of 95.21% and an F1 score of 95.18%. This model's superior performance can be attributed to the triplet loss function's efficacy in optimizing the feature space, providing clear separations between different activity classes.

Table 2 shows the comparison of the proposed SNN models with some of the well-known baseline models. The comparison illustrates the efficacy of Siamese networks, particularly with triplet loss, in handling the challenges of HAR. The SSN-TL model outperforms traditional and more recent approaches, highlighting the potential of leveraging structured loss functions and Siamese architectures for advanced activity recognition tasks.

**Table 2.** Comparison of proposed SSN model with well-known baseline models.

| Algorithm | Accuracy | F1 Score |
|-----------|----------|----------|
| LSTM | 90.8% | 90.8% |
| Bi-LSTM | 91.1% | 91.1% |
| Res-LSTM | 91.6% | 91.5% |
| ConvLSTM | 92.24% | NA |
| SSN-CL | 93.98% | 93.9% |
| SSN-TL | 95.21% | 95.18% |

*5.4. Analysis*

5.4.1. Critical Analysis on Loss Functions and Optimizers

The differential performance between the contrastive and triplet loss functions, as well as between the SGD and Adam optimizers, offers valuable insights.

A. Loss Function Impact:

The improved performance with the triplet loss function suggests that the additional context provided by the negative samples in triplet training offers a significant advantage in learning a more discriminative feature space for HAR tasks. This is particularly evident with the precision metric, where the triplet loss function with SGD outperforms the contrastive loss function, indicating a higher success rate in correctly identifying specific activities without increasing false positives.

Considering loss function impact, the triplet loss function shows a consistent improvement over contrastive loss across all metrics. For example, the accuracy improvement when moving from contrastive to triplet loss with SGD is about 1.23%, and, with Adam, it is only 0.72%. This pattern is observed across all metrics, indicating the triplet loss's effectiveness in enhancing model performance.

B. Optimizer Influence:

The consistent performance of the SGD optimizer across both loss functions highlights its suitability for the structured learning landscapes created by contrastive and triplet loss functions. While Adam is generally robust and adaptable, its performance in this context suggests that the adaptive learning rates might not always align optimally with the loss landscapes of HAR tasks, especially when using the triplet loss function.

While considering optimizer impact, within each loss function category, SGD tends to outperform Adam slightly. For instance, under contrastive loss, the difference in accuracy between SGD and Adam is 1.5%, and, under triplet loss, it is 2.01%. This trend is consistent across all metrics, suggesting that SGD may provide a more stable and effective optimization path for these specific loss functions in the given task.

5.4.2. Overall Evaluation:

It can be stated that the combination of triplet loss and SGD optimizer yields the best performance across all metrics, with an accuracy of 95.21%, precision of 95.19%, recall of 95.18%, and F1 score of 95.18%. This suggests that, for tasks similar to the one represented by these data, this combination might be the most effective.

The consistency in the ranking of configurations (triplet loss with SGD > triplet loss with Adam > contrastive loss with SGD > contrastive loss with Adam) across all metrics suggests that the observed performance differences are robust and not specific to a particular evaluation criterion.

The performance gaps between different configurations are relatively modest, particularly between the two optimizers for each loss function. This suggests that, while there are clear trends, the choice of optimizer might not be as critical as the choice of loss function for this specific task.

5.4.3. Performance Insights of Stacking Meta-Classifier

The stacking meta-classifier demonstrated a notable improvement in the overall classification accuracy and F1 scores, particularly when compared with individual base classifiers. This improvement can be attributed to several factors inherent in the stacking approach:

- Diversity of base classifiers: the stacking meta-classifier combines predictions from a variety of base classifiers, each with its unique strengths and biases. This diversity enables the meta-classifier to capture a broader range of patterns and relationships within the data, leading to more robust and accurate predictions.

- Complementary learning: different classifiers may excel in recognizing different types of activities or nuances within the HAR data. By aggregating their predictions, the stacking meta-classifier effectively harnesses their complementary capabilities, mitigating individual weaknesses and enhancing the overall predictive power.
- Weighted contributions: the final decision of the stacking meta-classifier is based on the weighted contributions of the base classifiers. This weighting mechanism allows the meta-classifier to prioritize more reliable predictions and adjust the influence of each base classifier based on its performance, further refining the classification outcomes.
- Error correction: the meta-classifier layer can serve as an error-correcting mechanism, effectively reconciling conflicting predictions from the base classifiers. This can be particularly beneficial in scenarios where certain activities are easily confusable, helping to reduce false positives and negatives.

## 6. Discussion

### 6.1. Challenges and Limitations of Siamese Neural Network Models

The experimental results from the Siamese neural network model, incorporating CNN, Bi-LSTM, and attention mechanisms for human activity recognition (HAR), alongside the utilization of contrastive and triplet loss functions, offer a rich dataset for analysis. This discussion delves into the impact of these components on HAR performance and evaluates the strengths and limitations of the chosen loss functions for embedding optimization.

#### 6.1.1. Impact of CNN, Bi-LSTM, and Attention Mechanisms
A. CNN Layers:

The convolutional neural network (CNN) layers play a critical role in spatial feature extraction from the sensor data. By capturing inherent spatial patterns related to different activities, CNN layers provide a foundational understanding of the data's physical characteristics. Their ability to hierarchically extract features makes them indispensable for distinguishing between activities with subtle spatial differences. However, CNNs primarily focus on spatial aspects and might overlook temporal dependencies crucial for HAR, where the sequence of movements is vital.

B. Bi-LSTM Layers:

The bidirectional long short-term memory (Bi-LSTM) layers address the temporal dimension that CNNs might neglect. By processing data both forwards and backwards, Bi-LSTMs capture temporal dependencies and the sequential nature of human activities, crucial for differentiating between activities with similar spatial features but distinct temporal patterns. The challenge with Bi-LSTMs, however, lies in their complexity and higher computational requirements, which can lead to longer training times and increased model complexity.

C. Attention Mechanisms:

The integration of attention mechanisms serves to enhance the model's focus on salient features, whether spatial or temporal. By weighting the importance of different features, the attention layer helps the network to prioritize critical information and improve classification accuracy. While highly effective, the success of attention mechanisms depends on their proper configuration and integration within the network, requiring careful tuning to avoid overshadowing other model components.

#### 6.1.2. Contrastive vs. Triplet Loss Functions
A. Contrastive Loss Function:

The use of a contrastive loss function, focusing on minimizing distances between similar pairs and maximizing distances between dissimilar pairs, is instrumental in creating a discriminative embedding space. This approach is straightforward and intuitive, making it a popular choice for tasks involving similarity learning. However, its binary nature

(comparing pairs) might limit the model's ability to capture the broader structure of the embedding space, potentially resulting in less optimal separation between different activity classes.

B. Triplet Loss Function:

The triplet loss function extends the pairwise comparison approach by incorporating an additional negative sample for each anchor-positive pair. This three-way comparison introduces a margin that further separates dissimilar activities, providing a more structured and well-separated embedding space. This can lead to superior performance in distinguishing between activities, as evidenced by the experimental results. The main limitation of the triplet loss function lies in its increased complexity and the challenge of selecting effective triplets, which can significantly impact training dynamics and convergence.

*6.2. Challenges of Stacking Meta-Classifier*

While the stacking meta-classifier has shown promising results, there are considerations and challenges to be mindful of:

- Model complexity: the addition of a meta-classifier layer increases the overall complexity of the model. This can have implications for training time, computational resources, and the risk of overfitting, necessitating careful regularization and validation practices.
- Hyperparameter tuning: the performance of the stacking meta-classifier is contingent on the selection of base classifiers and their hyperparameters, as well as the meta-classifier's own configuration. Optimal tuning is critical but can be time-consuming and computationally intensive, requiring extensive experimentation and cross-validation.
- Interpretability: the stacking approach, by virtue of combining multiple classifiers, can obscure the decision-making process, making it more challenging to interpret how specific predictions are derived. This may be a consideration in applications where transparency and explainability are important.

**7. Conclusions and Future Work**

This study has successfully demonstrated the utility of Siamese neural network architecture, enhanced with CNN, Bi-LSTM, and attention mechanisms, for the task of human activity recognition (HAR). The CNN layers effectively extracted spatial features, while the Bi-LSTM layers captured crucial temporal dependencies. Attention mechanisms further refined the model's focus on salient features, significantly improving activity classification accuracy. The application of contrastive and triplet loss functions for training the Siamese network optimized the embedding space, leading to superior discriminative capability. Additionally, the implementation of a stacking meta-classifier leveraged the strengths of multiple base classifiers, further enhancing the model's performance.

The results from this study affirm the efficacy of Siamese neural networks in HAR, particularly when combined with triplet learning and stacking meta-classifiers. This approach adeptly navigates the spatial–temporal complexities of human activities, offering a robust framework for accurate activity recognition and classification. The superior performance of SSN-CL and SSN-TL over well-known baseline models in HAR can be attributed to their sophisticated architecture that adeptly captures the complex spatial–temporal nature of human activities, the strategic optimization of embeddings through contrastive and triplet loss functions, and the enhanced decision-making facilitated by the stacking meta-classifier.

The promising outcomes of this study open several avenues for future research, which are as follows:

- Exploring semi-supervised learning: given the often limited availability of labeled data in HAR, applying semi-supervised learning within the Siamese network framework could be a fruitful direction. This approach could leverage unlabeled data to en-

hance model performance, making the training process more efficient and potentially improving the generalizability of the model.

- Transfer learning and domain adaptation: the application of transfer learning and domain adaptation techniques could address challenges related to dataset diversity and sensor discrepancies, enhancing the model's adaptability and reducing the reliance on extensive labeled datasets.
- Real-time and on-device deployment: optimizing the network for real-time processing on mobile or wearable devices presents a practical challenge. Future work could focus on model efficiency to facilitate deployment in real-world applications without sacrificing accuracy.
- Cross-domain applications: the proposed network architecture's versatility suggests potential applications beyond HAR, in domains requiring detailed classification or recognition tasks, such as gesture recognition, anomaly detection, or medical diagnostics.

Incorporating semi-supervised learning into the Siamese network for HAR represents a particularly compelling direction for future research. This approach could significantly expand the model's learning capacity, enabling it to leverage the vast amounts of unlabeled data often available in HAR scenarios. Such advancements could pave the way for more scalable, efficient, and adaptable HAR systems, with broad implications across various domains and real-world applications.

**Author Contributions:** Conceptualization, B.V. and B.-R.C.; methodology, B.V.; software, B.-R.C.; validation, B.V. and B.-R.C.; data curation, B.V.; writing—original draft preparation, B.-R.C.; writing—review and editing, B.V.; visualization, B.V.; funding acquisition, B.-R.C. All authors have read and agreed to the published version of the manuscript.

**Funding:** This study was supported by the Basic Science Research Program through the National Research Foundation of Korea, funded by the Ministry of Education (No. NRF-2017R1A6A1A03015496).

**Data Availability Statement:** Data are contained within the article

**Conflicts of Interest:** The authors declare no conflicts of interest.

## Abbreviations

The following abbreviations are used in this manuscript:

| | |
|---|---|
| HAR | Human activity recognition |
| CNN | Convolutional neural network |
| LSTM | Long short-term memory |
| Bi-LSTM | Bidirectional long short-term memory |
| SNN | Siamese neural network |
| CL | Contrastive loss |
| TL | Triplet loss |

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
