# Peer review of "Enhancing Human Activity Recognition with Siamese Networks: A Comparative Study of Contrastive and Triplet Learning Approaches"

_electronics, doi:10.3390/electronics13091739_

Round 1

Reviewer 1 Report

Comments and Suggestions for Authors

This paper explores the use of Siamese Neural Networks (SNNs) in the domian of Human Activity Recognition(HAR). It evaluates the effectiveness of Contrastive Loss Functio and Triplet Loss Funcation within the proposed network architecture. Overall, I think the experimental setup and subsequent analysis are logcial and reasonable, contributing to advancing research in HAR field.

I have some questions regarding the experiment design and evaluation as shown below. 

1. In Section 6.1.1, you discuss the influence of various layers. Have you conducted any comparative experiments with benchmark networks by modifying the architucture, such as ommiting certain layers? Such a benchmark analysis may further validate your observations about the impact of different layers. 

2. In Section 5.3.1, you mentioned 2 F1-scores, mean F1-score and weighted F1-score. However, the results section only references "F1 score" in Table 1 and Table 2. Could you please clarify which specific F1-score metric was used in these tables and explain the rationale behind selcting this particular F1-score for your analysis. 

Author Response

Dear Sir/Madam,

Please find our response as the attachment.

Thank you.

Reviewer 2 Report

Comments and Suggestions for Authors

This manuscript explores human activity recognition using Siamese Neural Networks, comparing the effectiveness of contrastive and triplet learning methods. The reviewer has some comments to improve the manuscript: 

  1. Please elaborate further on the significance of your study and elaborate what sets it apart from other studies in this field.
  2. Section 3.3, which discusses the attention mechanism, requires further elucidation. Specifically, there's a need for a detailed explanation of how the model depicted in Figure 3 operates, how it prioritizes and consolidates information. Additionally, please provide more information about each element serving as input (Q, K, V), elucidate the operations involved, and specify the exact purpose of each operation.
  3. Please explain the concept of weight sharing between identical subnetworks. Does sharing imply that these two subnetworks have identical values? Please provide a more detailed explanation.
  4. Add a section about the hyperparameters present in your proposed model and how you have tuned the model, as well as the figure of the loss function and the number of epochs.
  5. Equation 3 requires further elucidation. Kindly provide a detailed explanation of the parameters within the equation.
  6. The labels and numbers in Figures 7, 8, 9, and 10 are small. Please adjust the font size.
  7. The confusion matrix in Figure 9 indicates a significant disparity between Adam and SGD in the sitting parameter, while the differences in other parameters are minimal. Could you please clarify the reason for this?

Comments on the Quality of English Language

Minor English corrections are suggested.

Author Response

Dear Sir/Madam,

Please find our response in the attachment.

Thank you.

Round 2

Reviewer 2 Report

Comments and Suggestions for Authors

The manuscript has been improved and is ready for publication.